# Use of Ozone for Disinfection of PHARMODUCT^®^ Automatic System for Antineoplastic Compounding

**DOI:** 10.3390/ph18020140

**Published:** 2025-01-22

**Authors:** Vito Lovino, Antonio Riglietti, Anna Tolomeo, Giuseppe Capasso, Miriana Di Vittorio, Stefano Brattoli, Giuseppe Tesse, Vincenzo Dimiccoli, Marco Spartà, Luana Perioli

**Affiliations:** 1Itel Telecomunicazioni srl—Radiopharmaceutical Division, Via Antonio Labriola SNC, 70037 Ruvo di Puglia, Italy; a.riglietti@itelte.it (A.R.); a.tolomeo@itelte.it (A.T.); m.divittorio@itelte.it (M.D.V.); s.brattoli@itelte.it (S.B.); g.tesse@itelte.it (G.T.); v.dimiccoli@itelte.it (V.D.); 2Bioduct s.r.l., Via Di Collodi, 6/C, 50141 Firenze, Italy; marco.sparta@dedalus.eu; 3Department of Pharmaceutical Sciences, University of Perugia, Via del Liceo 1, 06123 Perugia, Italy; luana.perioli@unipg.it

**Keywords:** ATCC, ozone, automatic compounding, robot bio-decontamination, disinfection

## Abstract

**Background:** The purpose of this work was to demonstrate the ozone efficacy for disinfection of the PHARMODUCT^®^ automatic dispensing system for antineoplastic preparation, as a guarantee of a higher grade of cleanliness. While the use of ozone gas disinfection is almost consolidated in food and water treatment, there is a lack of scientific data in the pharmaceutical field. The scope of this study was to demonstrate the ozone efficacy for disinfection of the PHARMODUCT^®^ automatic dispensing system, before starting the antineoplastic preparation, in order to ensure a high degree of cleanliness and, at the same time, to define a biodecontamination procedure that could also be translatable to other automated compounding systems on the market. **Methods**: Ozone efficacy was determined by calculating the difference (pre-exposure–post-exposure) in CFU counts on the plate. A group of four different ATCC-selected microbial strains were tested using two distinct cycles. The first one was evaluated with an ozone gas concentration of 40 ppm for 40 min; the second cycle increased the concentration to 60 ppm for the same duration. **Results**: Results showed that exposure to 40 ppm ozone gas led to a 4-log reduction of all tested ATCC strains. In contrast, exposure to 60 ppm ensured a 6-log reduction. **Conclusions:** The ozone disinfection process, applied to the PHARMODUCT^®^ system, provides a superior grade of cleanliness compared to the manual disinfection procedure, thus offering insight beyond the current anti-inflammatory and analgesic application of ozone therapy in the medical field.

## 1. Introduction

A sterile environment is essential for reconstituting cytotoxic agents as sterile injectable drugs. It is critical to protect the finished product from physical, chemical, biological, and pyrogenic contaminants, just as it is equally important to minimize the exposure of health care workers as much as possible, since cytotoxic drugs also impact normal cells by inducing side effects such as teratogenicity, baldness, infertility and immunosuppression [1].

In order to preserve the degree of sterility of the environment in which injectable drugs are compounded, techniques currently applied in hospital settings involve the use of an isolator or vertical laminar airflow hood with a rigid surface or flexible film.

However, the cleaning and sanitization procedures applied to these aseptic environments still remain critical.

Several studies, conducted in pharmaceutical practice, have highlighted the importance of proper aseptic technique during the manipulation of injectables [2,3]. Potential risk factors involving either microbiological or particulate contamination may be the nature and complexity of the sterile drug compounding operation, the type of disinfectant used as well as its mode of application, and, finally, also the skill and level of training of the health care operator in current good manufacturing practices [2].

Chemical disinfectants and sanitizers, with or without detergents, are to date an effective method of ensuring surface hygiene in clinical and hospital areas [4].

However, adopting manual cleaning procedures, personnel become directly responsible, for example, for the type of dilution needed for that given disinfectant and that given surface on which it is to be applied, or for the order of application of the various disinfectants chosen, in order to avoid cross-contamination and ensure constant effectiveness, or for contact times on the various surfaces. Another non-negligible factor arising from the use of chemical disinfectants is the occupational health risk to the personnel using them [4].

The continuous search for low-temperature disinfection techniques arises from the need to ensure a certain compatibility between the sterilizing agent and the product’s physiochemical characteristics, to achieve faster processing times, and to meet environmental requirements. It is therefore important to consider the adoption of more environmentally friendly technologies and safer decontamination methods.

Therefore, in order to minimize the contribution of contamination in aseptic environments and obviate the limitations of chemical disinfectants, related to manual cleaning procedures, the PHARMODUCT^®^ automated system was developed (Figure 1), which allows automated preparation of antineoplastic agents and is equipped with an ozone-based bio-decontamination system, thus offering a higher degree of safety for personnel and grade A EU GMP work environment, with an EU GMP grade B background.

The use of ozone has become a viable alternative to common sterilization methods, such as gamma irradiation, ultraviolet irradiation, dry and moist heat, steam, hydrogen peroxide (gas and liquid), and ethylene oxide [5].

Ozone gas, a triatomic oxygen molecule, presents several potential advantages over other decontamination gases and liquid chemical applications.

Ozone is a safer alternative to Ethylene oxide (EtO): EtO is flammable, explosive and has been recognized as potentially mutagenic to workers, such that accidental exposure to EtO can pose a significant safety risk.

Ozone is also preferable to UV disinfection because ultraviolet irradiation could cause burns, eye damage or increase the risk of skin cancer, due to overexposure. Additionally, it is known that microorganisms can grow in crevices and UV light cannot completely penetrate hidden areas.

On the other hand, ozone inhalation may irritate the respiratory tract and cause lung damage, with symptoms such as coughing and chest tightness commonly associated with prolonged exposure [6]. However, the health hazards can be overcome in practice by ensuring that the room to be treated is temporarily closed during the treatment and it is sealed to prevent the gas from escaping into the environment by means of an airtight neoprene gasket.

Ozone is significantly less stable than atmospheric oxygen [7]; this instability means that it does not generate accumulation phenomena and must be produced only when needed, at the point of use, by means of a suitable ozone generation system. Ozone automatically and rapidly decomposes into oxygen in both air and aqueous environments, with a high oxidation potential of 2.07 V, making it effective against a broad spectrum of microbial species [5,8].

Gas can penetrate all areas including crevices, fixtures, fabrics, and undersurfaces of furniture, more efficiently than the manually applied liquid spray or aerosols.

Moreover, it is remarkably simple and inexpensive to produce. Ozone can be generated by UV radiation, an electrochemical process and corona discharge.

However, if the first two methods lead to reduced ozone concentration and low regulation of the process [9], in the system defined as the corona discharge ozonator, ozone is formed when oxygen passes through a gap between high-voltage and ground electrodes in order to create an energy field, called a corona.

The energy of the electrical discharge allows the splitting of oxygen molecules into single oxygen atoms until the three-atom ozone molecule is formed; this causes the energy released by the electrical discharge, necessary for the formation of ozone, to be released from the ozone itself upon its decomposition, resulting in its high sterilizing effectiveness.

In addition, the PHARMODUCT^®^ device is equipped with one or more carulite chemical scavengers; this can quickly reduce the ozone concentration inside the chamber without releasing significant residual oxidants into the drain.

Ozone, therefore, due to its high oxidizing capacity, is now widely used in various industries, such as food or water treatment, as it can clean, disinfect and sterilize materials and surfaces, depending on the medium used and the application dosage [5].

In the health and biomedical fields, ozone is currently used to sterilize medical devices, particularly those that are sensitive to high temperatures. However, a comprehensive analysis of the close correlation between ozone concentration and exposure time, for different classes of bacteria and yeasts, applied for disinfection of surfaces and environments in the preparation of antineoplastic drugs is lacking in the literature.

Therefore, the purpose of this study is to evaluate the efficacy of the automated ozone bio-decontamination process applied to the PHARMODUCT^®^ automatic dispensing system, which enables the automated production of multi-dose bags and final preparations of antiblastic drugs in a safe mode for the operator and the preparation itself [1].

## 2. Results

In our study, we evaluated the PHARMODUCT^®^ equipment, using four representative ATCC strains (*E. coli* ATCC 25922, *S. simulans* ATCC 27848, *B. cereus* ATCC 11778, *S. kudriavzevii* ATCC 2601), positioned in the worst-case point. We aimed to evaluate two different effects of ozone treatments.

Both tests maintained the same exposure time; however, we aimed for a 6-log reduction of all strains using 60 ppm of ozone and a 4-log reduction using 40 ppm of ozone to ensure a satisfactory level of disinfection (at least 10^4^ CFU of vegetative forms) [10].

The ozone disinfection process in the PHARMODUCT^®^ automated system was effective against all ATCC microbial strains tested, satisfying all predefined acceptance criteria:■All exposed plates, including the negative control, showed no microbial growth.■The positive control confirmed stain vitality and theoretical titer.

Specifically, in the main chamber, a cycle with an ozone concentration of 40 ppm for 40 min, carried out in triplicate for each position and for all four strains for each run with the same analytical conditions, was able to eliminate a load of 10^4^ microorganisms at each test point (Table 1).

When the concentration was increased to 60 ppm for the same duration, the system successfully killed 10^6^ microorganisms per point, in the same analytical conditions as the previous test (triplicate for each position and strain), resulting in a 2-log increase in the killing efficiency (Table 2).

## 3. Discussion

Although numerous laboratory-scale studies have explored the effects of ozone inactivation on various organisms and contaminants, few reports are available on its industrial application as a disinfection method. While extensive research has been conducted on the use of ozone in the food industry and drinking water treatment, fewer studies have focused on its application in pharmaceuticals.

To enable the widespread use of ozone disinfection technology, automated systems that can reduce cycle time and provide rapid disinfection are essential.

Table 3 presents a summary of some of the advantages and disadvantages of using ozone compared to manual disinfection techniques [11], which make use of chemical disinfectants, such as, for example:-chlorine derivatives, which although cheap and having good microbicidal activity, inactivate more slowly than ozone and are corrosive to metal surfaces;-quaternary ammonium salts, which have good cleaning properties and are easy to handle, although they can cause irritation of the respiratory tract and skin up to severe caustic burns on the skin and in the gastrointestinal tract (depending on the concentration) or other gastrointestinal symptoms (e.g., nausea and vomiting) or coma, convulsions, hypotension and even death;-alcohols, which, although they have broad-spectrum antiseptic properties, are potentially flammable and irritating and toxic after prolonged contact with the skin.

In general, ozone disinfection overcomes most of the critical issues related to the manual disinfection process.

With just 40 ppm of ozone, 4-log microbial reduction disinfection can be achieved. An additional 20 ppm results in 6-log reduction disinfection against common bacteria and yeasts, all at low production costs. Only two electric hobs are needed to produce ozone using environmental air as the raw material.

Furthermore, the high oxidant power of ozone offers the possibility of effective results against other various microbial strains and resistant spores [12,13,14,15].

## 4. Materials and Methods

### 4.1. Materials

TSA 90 mm plates were purchased from Merck KGaA (Darmstadt, Germany), SDA 90 mm plates were purchased from Merck KGaA (Darmstadt, Germany), NaCl Peptone buffer pH 7.0 was purchased from Merck KGaA (Darmstadt, Germany), Frankfurter Str. 250, 64293 Darmstadt, Germany. *Escherichia coli* ATCC 25922, *Staphylococcus simulans* ATCC 27848, *Bacillus cereus* ATCC 11778, *Saccharomyces kudriavzevii* ATCC 2601 were purchased from LGC Standards S.r.l., Via G. Carducci, 39, 20099 Sesto San Giovanni, MI, Italy.

### 4.2. PHARMODUCT^®^ System Description

The PHARMODUCT^®^ is a medical device designed for use in hospital oncology wards. It automates the preparation of multidose bags and final preparations of antiblastic drugs, which can be dispensed in bags, syringes and elastomer pumps. It is provided with precision scales, monitoring the accuracy of the product weight to be set up, peristaltic pumps used to transfer liquids, a dissolution station for powders and isolated waste areas with thermally sealed bags, in which to confer the residual contents of antineoplastic drugs, effectively preventing any contamination of the work chamber [1].

The dispensing automated system is installed in an isolated compartment consisting of a negative-pressure laminar flow chamber with ULPA-U15 filters, classified as ISO class 5, according to EN ISO 14644-1 [16].

A system of inflatable gaskets surrounds the chamber openings, ensuring perfect airtightness, also in the event of pneumatic and electrical power failures, allowing the panel to be moved easily and safely, without any locking systems, and also ensuring a perfect seal against the external environmental grade B EU GMP background.

It is also equipped with a unique ozone vapor-based decontamination system that ensures the achievement of specific ppm ozone concentrations validated during the system’s qualified performance.

This ensures the maintenance of unaltered EU GMP Grade A microbiological and particulate conditions, in accordance with the requirements to the cGMP guidelines [17] for the manufacture of sterile products.

### 4.3. Ozone Chamber

The PHARMODUCT^®^ features a system designed to produce artificial ozone by applying the corona effect; adjusting both the concentration in ppm and the ozone exposure time according to the settings defined by the end user, the machine generates an electric voltage that produces negative ions and ozone for air decontamination. Internally, there are two ceramic powered plates at a high voltage of 220 V for ozone generation and the equipment incorporates an ozone sensor, an electrochemical device used to detect O_3_ gas.

O_3_ sensors operate by generating a small electrical current, proportional to the partial pressure of ozone gas in the surrounding air. The current is the result of ozone reduction on the surface of a catalytic electrode (according to the reaction below), with a resulting signal that is linear with ozone concentration.O_3_ + 2H^+^ + 2e^−^ ⇄ O_2_ + H_2_O

### 4.4. Mechanisms of Ozone Action

Ozone destroys microorganisms by oxidizing vital cellular components. The bacterial cell surface has been suggested to be the primary target of ozonation. Two major mechanisms in the destruction of the target organisms by ozone have been identified. The first mechanism is that ozone oxidizes the sulfhydryl groups and amino acids of enzymes, peptides and proteins to shorter peptides, while the second mechanism is that ozone oxidizes polyunsaturated fatty acids to acid peroxides, with consequent bacterial cell death [18].

Besides the direct action of ozone, microbial inactivation by ozone may also be indirect. This indirect inactivation results in the effect of reactive oxidative species (ROS including OH^●^, HO_2_^●^, O_2_^●^, O_3_^●^, HO_3_^●^, H_2_O_2_, O), which may be produced during spontaneous ozone decomposition in water or air. The hydroxyl radical (OH^●^), for example, is highly unstable, reacts readily with other cytoplasmic component to gain the missing electron and has a higher oxidation potential (2.80 V) compared to ozone itself (2.07 V) [5].

### 4.5. Methods

Four microbial strains commonly found in humans and the environment [5,19,20,21,22] were selected as representatives of microbial contamination:*Escherichia coli* ATCC 25922;*Staphylococcus simulans* ATCC 27848;*Bacillus cereus* ATCC 11778;*Saccharomyces kudriavzevii* ATCC 2601.

Two types of ozone disinfection cycles were validated in the PHARMODUCT^®^ machine. The first cycle involved 60 ppm of ozone for 40 min of exposition, resulting in a 6-log reduction of each ATCC strain. The second cycle provided 40 ppm for 40 min, achieving a 4-log reduction.

#### 4.5.1. 6-Log Test

The first test involved exposing 90 mm plates containing a population of ≥10^6^ for each ATCC microbial strain (one strain per plate) within the PHARMODUCT^®^ system.

Five different critical positions were tested for each microbial strain. These positions were selected based on the areas of greatest potential contamination, including the working area where the mechanical components of the machine are in motion, the needle penetration site, the liquid transfer point and the powder reconstitution area.

A “negative control” plate, which was not inoculated, was set aside. Two plates for each strain were inoculated and were not exposed to the ozone cycle; these serve as “positive controls” for titer checks.

The ozone disinfection cycle was programmed on the machine to operate at a concentration of 60 ppm for 40 min of exposure.

At the end of the cycle, all plates (positive control included) were incubated at 30–35 °C for 24 h (bacteria) and at 20–25 °C for 48–72 h (yeasts). The negative control was incubated at 30–35 °C for 72 h.

The analytical test, 6-log Test, was repeated three times under the same analytical conditions as previously stated.

#### 4.5.2. 4-Log Test

The test was conducted by exposing 90 mm plates containing a population ≥10^4^ ATCC strains within the PHARMODUCT^®^ system. The mode of execution of the operational test follows the same methods used in the “6-log Test”.

The same five critical positions previously established were tested for each microbial strain.

A “negative control” plate, which was not inoculated, was set aside. Two plates for each strain were inoculated and were not exposed to the ozone cycle; these serve as “positive controls” for titer checks.

The ozone disinfection cycle was programmed on the machine to operate at a concentration of 40 ppm for 40 min of exposure.

At the end of the cycle, all plates (positive control included) were incubated at 30–35 °C for 24 h (bacteria) and at 20–25 °C for 48–72 h (yeasts). The negative control was incubated at 30–35 °C for 72 h.

The analytical test, 4-log Test, was repeated three times under the same analytical conditions as previously stated.

#### 4.5.3. Execution Plan

1.Preparation of strain suspensions

In this phase, a “concentrated microbial suspension” was prepared for each strain to be tested. Each suspension had a concentration of ≥1.5 × 10^8^ CFU, equivalent to 0.5 on the standard McFarland scale. The preparation process involved the following steps:(a)A fresh crop was grown on a TSA plate starting from the Master Cell Bank of each strain.(b)Next, a bulk of microorganisms was re-suspended in 5 mL of NaCl Pepton Buffer pH 7.0 to achieve turbidity comparable to 0.5 on the McFarland scale.

The titer of the “concentrated suspension” was verified by performing serial dilutions with NaCl Pepton diluent buffer to achieve a theoretical concentration of 10^3^ CFU/mL:

1° dilution (A) → 1 mL of (concentrated suspension) + 9 mL of diluent (dilution 1:10)

► titer obtained~1.5 × 10^7^ cells/mL;

2° dilution (B) → 1 mL of (A) + 9 mL of diluent (dilution 1:10)

► titer obtained~1.5 × 10^6^ cells/mL;

3° dilution (C) → 1 mL of (B) + 9 mL of diluent (dilution 1:10)

► titer obtained~1.5 × 10^5^ cells/mL;

4° dilution (D) → 1 mL of (C) + 9 mL of diluent (dilution 1:10)

► titer obtained~1.5 × 10^4^ cells/mL;

5° dilution (E) → 1 mL of (D) + 14 mL of diluent (dilution 1:15)

► titer obtained~10^3^ cells/mL.

To verify the titer of the “concentrated suspension”, 100 µL of “suspension E” was seeded onto two TSA plates (100 µL for each plate) to obtain approximately 10^2^ cells/plate. Therefore, the expected count was approximately 100 UFC per plate. All the plates were incubated at 30–35 °C for 24 h (bacteria) and 20–25 °C for 48–72 h (yeasts).

Compliance was considered achieved if the average counts on the two plates after incubation were ≥100 UFC, confirming the theoretical data of suspension titer.

2.Preparation of “test suspensions”

All plates intended for exposure to the ozone cycle were prepared and their microbial reduction was assessed after the cycle.

For each position to be tested inside the PHARMODUCT^®^ machine, one plate of TSA or SDA (depending on the microbial strain) was inoculated with the corresponding “Test suspension” and exposed to the ozone cycle. In the first set of plates, the concentration of “Test Suspension No. 1”, for each strain, was 10^6^ CFU or more. The second set of plates used “Test Suspension No. 2” with a concentration of 10^4^ CFU or more. Both concentrations were achieved by appropriately diluting the “concentrated suspensions” mentioned in the previous point 1.

For the choice of parameters tested, regarding the concentration of the microbes, as per European Pharmacopoeia [23], the SAL, sterility assurance level, expressed as the probability of survival of microorganisms in a given environment and/or product, after exposure to the sterilization/biodecontamination process, was used as a reference.

During physical and biological validation of the biodecontamination cycle within the PHARMODUCT^®^ system, the physical conditions were set to meet a SAL ≤ 10^−6^, thus demonstrating the efficacy of the process at multiple locations within the automated system’s inner chamber.

In addition, complete inactivation (Total kill Analysis) of a 6-log population of representative bioburden is sufficient to demonstrate an appropriately low risk of contamination in an isolator used for aseptic operations [24].

Instead, according to UNI ISO 13697 [25], a specific product to be defined as a chemical disinfectant with bactericidal and/or fungicidal and/or yeasticidal activity must demonstrate at least a 4-log decimal reduction in viability for bacteria and at least a 3-log decimal reduction in viability for fungi.

About ozone concentration, although the disinfectant action of ozone can also be achieved at lower ppm values and exposure times than those tested, as reported in the literature, it was preferred to test a minimum concentration of 40 ppm for 40 min, as a worst case, in order to demonstrate the total inactivation of contaminants placed inside the work chamber. This was possible considering zero risk of critical and prolonged exposure as a resultant effect on humans, since the automated compounding system of antineoplastic drugs, isolated from the external environment with seals, following the ozonation phase, also includes an ozone suppression phase using chemical carulite filters, thus ensuring operation in maximum safety.

Therefore, it is possible to consider that the evaluation criteria used to perform this validation study and the choice of microbial concentrations tested are likely to be representative for the intended applications of the PHARMODUCT^®^ system.

These preparations were prepared as follows. Starting from a concentrated suspension at 0.5 McFarland, equal to 1.5 × 10^8^ CFU/mL, a 1:10 dilution was performed to obtain “Test Suspension No. 1” with a concentration of approximately 1.5 × 10^7^ CFU/mL. Additionally, from the same concentrated suspension at 0.5 McFarland, three serial dilutions 1:10 were performed to obtain “Test suspension No. 2” with a concentration of 1.5 × 10^5^.

The first set of 6-log reduction test plates was inoculated with 100 μL of “Test Suspension No. 1” to obtain a theoretical microbial inoculum concentration of 1.5 × 10^6^ CFU/plate. Similarly, the second set of plates for the 4-log reduction test was prepared using the same procedure, inoculating 100 μL of “Test Suspension No. 2” to obtain a microbial concentration of 1.5 × 10^4^ CFU/plate.

3.Positive control/titer verification

This process allows us to verify the viability of the microorganisms and to evaluate the titer of each test suspension.

Two plates were prepared for each strain without exposure to the ozone-cycle. Starting from “Test Suspension No. 1”, appropriate serial dilutions were performed to obtain a “Check Suspension No. 1” with a titer of ≥10^3^ CFU/mL. Then, 100 µL of “Check Suspension No. 1” was inoculated onto each plate for each microbial strain, aiming for a theoretical microbial inoculum concentration of 10^2^ CFU/plate. This procedure was then repeated for “Test suspension No. 2” and “Control Suspension No. 2”.

All the plates were properly incubated at 32.5 °C (for bacteria) and 22.5 °C (for yeasts) until growth appeared within 3 days (for bacteria) and 5–7 days (for yeasts).

4.Negative controls

The purpose of these controls is to verify that sources of contamination unrelated to planned activities—whether from operators, the environment or other factors—do not affect test results. The negative control plates (TSA and SDA) were not inoculated or exposed to ozone, but were handled in the same way as the other plates.

5.Loading the PHARMODUCT^®^ main chamber and positioning of the test plates

The main chamber of PHARMODUCT^®^ was free of any unnecessary materials and had been cleaned according to the established procedures.

The first set of six plates was designated for specific locations (Figure 2), with one plate allocated for each microbial strain at every point. Each plate contained a concentration of 10^6^ CFU per plate:(a)Upper rotating plate unit;(b)Peristaltic pump assembly;(c)Multi-dose bag scale plate;(d)Lower rotating plate unit;(e)Chamber lower surface.
6.Starting of ozone cycle

The ozone cycle was set to 60 ppm and run for 40 min.

Once the cycle was completed, the main chamber was opened and all plates were closed and placed in a sterile container.

A similar procedure was followed for plates treated with a 4-log setting, with the ozone cycle adjusted to 40 ppm for 40 min.

7.Incubation of plates

All plates tested were incubated for 24–48 h at 32.5 ± 2.5 °C for bacteria (TSA) and 48 h–72 h at 22.5 ± 2.5 °C for yeasts (SDA). Negative controls were incubated at 32.5 °C for 3 days followed by another 2 days on 22.5 °C.

8.Plate reading

At the end of the incubation period, ITELPHARMA’s microbiological laboratory visually inspected all plates and counted the microbial colonies, verifying, in accordance with the acceptance criteria, that:Negative controls must not show any bacterial or yeast growth;Positive controls should show growth ≥ 10^3^ CFU/mL, confirming flat titer;Plates treated with two ozonation cycles—one at 60 ppm for 40 min and another at 40 ppm for 40 min—should demonstrate a reduction of 6-log and 4-log, respectively. To confirm the effectiveness of the treatment, the plates for each strain should show no growth, meaning no colonies should be present.

## 5. Conclusions

This study evaluates the potential of high-level ozone as a disinfectant, thus going beyond the current anti-inflammatory and analgesic application of ozone therapy in the medical field.

Any application must ensure that human exposure to unsafe levels of ozone is prevented. Specifically, the research analyzes the effects of high levels of ozone on two varieties of bacteria and two types of yeast.

The impact of ozone on test organisms was assessed by comparing the number of CFUs recovered from the exposed surfaces, such as slides or wallboard, with those recovered from the unexposed test surfaces. The ozone cycle used in the PHARMODUCT^®^ automated system provides efficient and reliable disinfection, providing numerous advantages over traditional manual disinfection methods, including excellent compatibility with a wide range of materials and several polymers commonly used in pharmaceuticals, as well as reasonable costs for its use [11,26,27].

The ability to cover the entire surface, including hard-to-reach areas, minimizes the risk of overlooking impervious surfaces or corners within the cytotoxic drug preparation chamber. As a result, this method ensures a higher level of disinfection compared to manual approaches.

This biodecontamination process could be a key and critical point in hospital environments for the preparation of injectable antineoplastic drugs, but most of all, it represents an interesting opportunity, as it is also applicable to other automated compounding systems currently on the market.

## Figures and Tables

**Figure 1 pharmaceuticals-18-00140-f001:**
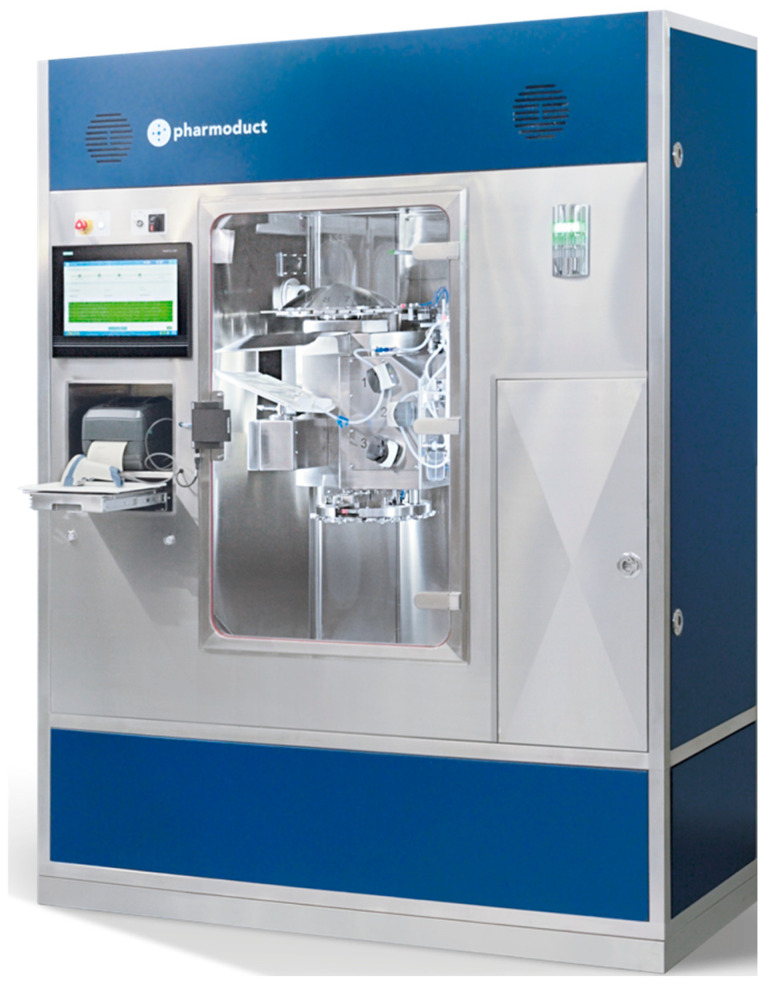
The outside of the PHARMODUCT^®^ automatic system built by Bioduct s.r.l. Firenze 50141, Italy, which is used for antineoplastic preparation.

**Figure 2 pharmaceuticals-18-00140-f002:**
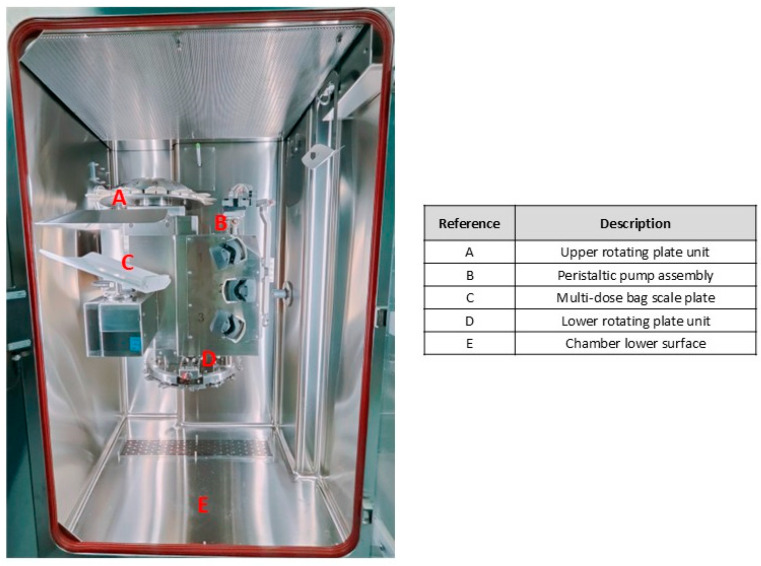
Layout of PHARMODUCT^®^ main chamber with plate positions.

**Table 1 pharmaceuticals-18-00140-t001:** Results of tests conducted starting from an initial microbial concentration of ≥10^4^.

4-Log Test				
Microbial strain	N°. Run	Initial population	Final population	Deviation standard
*Escherichia coli* ATCC 25922; *Staphylococcus simulans* ATCC 27848; *Bacillus cereus* ATCC 11778; *Saccharomyces kudriavzevii* ATCC 2601.	1	≥10^4^	0 (zero)	0 (zero)
	2	≥10^4^	0 (zero)	
	3	≥10^4^	0 (zero)	

**Table 2 pharmaceuticals-18-00140-t002:** Results of tests conducted starting from an initial microbial concentration of ≥10^6^.

6-Log Test				
Microbial strain	N°. Run	Initial population	Final population	Deviation standard
*Escherichia coli* ATCC 25922 *Staphylococcus simulans* ATCC 27848 *Bacillus cereus* ATCC 11778 *Saccharomyces kudriavzevii* ATCC 2601	1	≥10^6^	0 (zero)	0 (zero)
	2	≥10^6^	0 (zero)	
	3	≥10^6^	0 (zero)	

**Table 3 pharmaceuticals-18-00140-t003:** Some advantages and disadvantages of the manual disinfection method compared to the use of ozone.

Disinfection Method	Advantages	Disadvantages
Manual disinfection	Continuity of biodecontamination activities without incurring instrumental malfunctions	Repeatability of the method limited by human factors like experience, skill, training and physical and mental health
		Risk of microbial contamination due to human contact during manual operations
	Possibility to use any disinfectant/sanitizing agents validated by the end user	Duration of operations can be lengthy
		Risk of chemical contamination from disinfectant residuals
		Properly disinfecting small or hard-to-reach surfaces and components may be challenging
Ozone	Flexible application (variable efficacy depending on concentration, duration, and range of application)	Toxic gas if inhaled in relatively small amounts can cause chest pain, burning eyes and mucous membrane irritation followed by migraines; long-term chronic exposure can be fatal
	Environmentally friendly (decomposes to O_2_)	
	Excellent penetration into hard-to-reach areas of an object	
	Repeatability of the method ensured by establishing a standardized and automated process	
	No risk of human contamination since no people are involved in the process	Ozone utilization is usually influenced by system design and operating conditions. Not all added ozone can be effectively utilized, as it naturally decomposes during the reaction process.
	Duration of the ozone disinfection process faster than the manual methods	
	No risk of residual substances, as the only by-product of ozone degradation is oxygen	
	Its ability to diffuse as a gas ensures disinfection of all surfaces, including air sanitation, which is not typically achieved with manual disinfection.	
	Fairly low installation and operating costs	

## Data Availability

The original contributions presented in this study are included in the article. Further inquiries can be directed to the corresponding authors.

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
