# Peer review of "Use of Ozone for Disinfection of PHARMODUCT® Automatic System for Antineoplastic Compounding"

_pharmaceuticals, 2025, doi:10.3390/ph18020140_

Round 1

Reviewer 1 Report

Comments and Suggestions for Authors

The study aimed to prove the efficacy of ozone as a disinfectant applied in a commercial product for automatic compounding of antineoplastic drugs. In my opinion it is an useful study that can add knowledge of the use of ozone in disinfecting automated compounding systems. But there are a few things that needs to be addressed.

I have some general remarks and specific questions listed below.

Abstract: Unless the journal accepts the abbreviations as generally accepted, please explain the abbreviations CFU and ATCC in the abstract as well the first time it is used in the main text.

Materials and methods section:

Paragraph 2.2 and 2.3 System description: It is not specified if the system can be operated regarding the ozone concentration using the selected criteria only or if there is a continuous selection of settings that can be used. Please specify.

Please, describe the rationale of choosing the parameters for testing, both regarding the disinfection cycles (concentrations of the ozone and time frame) and the concentration of the microbes. Is the selected microbe concentration a “worst case” selection or chosen as a representative amount normally present in pharmaceutical preparation facilities? Why were the two settings chosen? Would it be more useful to assess and determine the number of non-viable microbe per ppm ozone and time used?

The outline of the tests described in paragraphs 2.5.1 and 2.5.2 are insufficiently described, see below.

Page 5, line 180 Please clarify the experiment so that the reader can easily follow the outline. The outline of the test is unclear, were there two kinds of test outlines, with different amounts of microbial content (106 and 104 CFU per plates respectively) and then each outline was tested with the two disinfectant cycles, either 40 or 60 ppm ozone?

Page 5, line 196 Please clarify the experiment. See comment above. Were 5 different positions also tested using the lower microbe concentration?

Results section:

Please state clearly the validation parameters (efficacy e.g.) and also number of runs per experiment type. Tabulated result would be helpful in order to get an overview of the results. Include the microbial reduction in each experiment and the standard deviation. It could for example include data of the experiment such as number of experiments (N=X) and also an estimation of the uncertainty. How many replicates/times was the experiment repeated for each position and strain?

If the concentration of each strain was 106 CFU per plate, then write the exact number of the read-out after disinfection. If it was zero, write that in the table. If the result was zero even after the 104-experiment, why did you not repeat experiments to assess even lower level of ozone-treatment?

Page 5, line 182 and 197, why is the concentration of the strain suspension set to >106 and >104? Was it the lower level that was critical?

In the discussion section, pleas address the following things:

In the discussion there is a list of disadvantages using manual disinfectant routines as well as a list of advantages using the automated ozone disinfection, but could you also address if there aren’t any advantages using the manual disinfection and also any disadvantage using ozone that could be useful for the reader to know about?

In the discussion, please add a reflection of the degree of microbial reduction that was observed in this study is in relation to expected levels of microbes in environments that the system is likely to be used in. To conclude, are the parameters used for validation, representative for the expected applications?

Also add a comment on how in a system like the one described in this study, using disinfectant gas for the reduction and elimination of microorganisms, how is the remaining content of antineoplastic drugs in the compounding system handled? Are there remaining levels of ADs in the compartment, and if so, how are these contaminants reduced or cleaned?

Comment the limitations of the study, such as low number of replicates.

Page 8, line 355 4-log microbial “reduction “ is missing.

Author Response

The study aimed to prove the efficacy of ozone as a disinfectant applied in a commercial product for automatic compounding of antineoplastic drugs. In my opinion it is an useful study that can add knowledge of the use of ozone in disinfecting automated compounding systems. But there are a few things that needs to be addressed.

I have some general remarks and specific questions listed below.

Comment 1 - Abstract: Unless the journal accepts the abbreviations as generally accepted, please explain the abbreviations CFU and ATCC in the abstract as well the first time it is used in the main text.

Response 1: Thank you for pointing this out. We have added the abbreviations CFU and ATCC in the Abstract (page 1 of 13, Title of paper, line 2) and we have added Paragraph 6 “Abbreviations” (page 11 of 13, Paragraph 6, line 451).

Comment 2: Materials and methods section. Paragraph 2.2 and 2.3 System description: It is not specified if the system can be operated regarding the ozone concentration using the selected criteria only or if there is a continuous selection of settings that can be used. Please specify.

Response 2: Thank you for pointing this out. We agree with this comment. Therefore, we have mentioned this in page 6 of 13, Paragraph 4, line 210-211.

Comment 3. Please, describe the rationale of choosing the parameters for testing, both regarding the disinfection cycles (concentrations of the ozone and time frame) and the concentration of the microbes. Is the selected microbe concentration a “worst case” selection or chosen as a representative amount normally present in pharmaceutical preparation facilities? Why were the two settings chosen? Would it be more useful to assess and determine the number of non-viable microbe per ppm ozone and time used?

Response 3: Thank you for pointing this out. We agree with this comment. Therefore, we have specified the choose of the concentration of the tested microbes in pages 8-9 of 13, paragraph 4.5.3, point 2 lines 320-334: the concentration of 106 was chosen according to paragraph 5.1.2 Pharmacopeia and PDA – TR-34; while the concentration of 104 was chosen in according to UNI EN 13697.

About the choice of the concentration of the tested ozone, we have specified in the main text on pages 9 of 13, paragraph 4.5.3, point 2 lines 335-343.

Comment 4. The outline of the tests described in paragraphs 2.5.1 and 2.5.2 are insufficiently described, see below.

Page 5, line 180 Please clarify the experiment so that the reader can easily follow the outline. The outline of the test is unclear, were there two kinds of test outlines, with different amounts of microbial content (106 and 104 CFU per plates respectively) and then each outline was tested with the two disinfectant cycles, either 40 or 60 ppm ozone?

Response 4:

Thank you for pointing this out. We agree with this comment. Therefore, we revised the two paragraphs for the two tests 6-log Test and 4-log Test. In particular, we have detailed the two tests, including a clear description of how they were performed. The two microbial concentrations (106 CFU per plate and 104 CFU per plate) were tested with 60 ppm and 40 ppm, respectively, but each individual microbial concentration was not tested with both ozone cycles.

We detailed two tests on page 7-8, paragraph 4.5.1-4.5.2, line 244-277.

Comment 5. Page 5, line 196 Please clarify the experiment. See comment above. Were 5 different positions also tested using the lower microbe concentration?

Response 5: Thank you for pointing this out. We agree with this comment. Therefore, we revised this paragraph for 4-log Test. In particular, we have detailed this test, including a clear description of how it was executed. We detailed this test on page 7-8 of 13, in paragraph 4.5.2, line 262-277. The five critical different positions were tested with plates containing a population of ≥10^4 ATCC strains, for each microbial strain.

Comment 6. Results section. Please state clearly the validation parameters (efficacy e.g.) and also number of runs per experiment type. Tabulated result would be helpful in order to get an overview of the results. Include the microbial reduction in each experiment and the standard deviation. It could for example include data of the experiment such as number of experiments (N=X) and also an estimation of the uncertainty. How many replicates/times was the experiment repeated for each position and strain?

Response 6: Thank you for pointing this out. We agree with this comment. Therefore, we entered two tables, named Table 1, for the 4-log Test, and Table 2, for the 6-log Test. Within the tables were included the results of tests conducted from an initial microbial concentration of ≥10^4, for Table 1, and the results of tests conducted from an initial microbial concentration of ≥ 10^6, for Table 2. Tables provide data for microbial reduction for each run and standard deviation data. The 4-log test was performed 3 times. The 5 positions were considered for each test, and the 4 plates were placed in each position. The 6-log test follows the same pattern as the 4-log test. Tables are inserted on page 4 of 13, paragraph 2, line 142 for table 1, line 148 for table 2.

Comment 7. If the concentration of each strain was 10CFU per plate, then write the exact number of the read-out after disinfection. If it was zero, write that in the table. If the result was zero even after the 104-experiment, why did you not repeat experiments to assess even lower level of ozone-treatment?

Response 7: Thank you for pointing this out. We agree with this comment. Therefore, the result obtained after reading the plates was zero (page 4 of 13, paragraph 2, lines 142-148). Ozone levels lower than 40 ppm were not tested, as the starting ozone concentration was chosen as the worst case, to be sure to inactivate any contaminants present inside the chamber.

Comment 8. Page 5, line 182 and 197, why is the concentration of the strain suspension set to >106 and >104? Was it the lower level that was critical?

Response 8: Thank you for pointing this out. We agree with this comment. Therefore, we defined this aspect in the response 3. We specified the choose of the concentration of the tested microbes in paragraph 4.5.3 point 2 lines 320-334: the concentration of 10^6 was chosen in according paragraph 5.1.2 Pharmacopeia and PDA – tr-34; while the concentration of 10^4 was chosen in according to UNI EN 13697.

Comment 9. In the discussion section, please address the following things:

In the discussion there is a list of disadvantages using manual disinfectant routines as well as a list of advantages using the automated ozone disinfection, but could you also address if there aren’t any advantages using the manual disinfection and also any disadvantage using ozone that could be useful for the reader to know about?

Response 9: Thank you for pointing this out. We agree with this comment. Therefore, We have analysed in more detail the advantages and disadvantages of manual cleaning compared to the use of ozone. This comparison is shown in a summary table (page 5 of 13, paragraph 3, lines 170-173), in which some advantages and disadvantages of the manual disinfection method are compared to the use of ozone.

Comment 10. In the discussion, please add a reflection of the degree of microbial reduction that was observed in this study is in relation to expected levels of microbes in environments that the system is likely to be used in. To conclude, are the parameters used for validation, representative for the expected applications?

Response 10: Thank you for pointing this out. We agree with this comment. Therefore, we have included a precise evaluation in page 8 of 13, paragraph 4.5.3, lines 320 - 346.

Comment 11. Also add a comment on how in a system like the one described in this study, using disinfectant gas for the reduction and elimination of microorganisms, how is the remaining content of antineoplastic drugs in the compounding system handled? Are there remaining levels of ADs in the compartment, and if so, how are these contaminants reduced or cleaned?

Response 11: Thank you for pointing this out. We agree with this comment. Therefore, we have added in section 4.2., the description of the equipment PHARMODUCT® system was expanded, specifying the presence of isolated, distinct areas for the delivery of the residual content of antineoplastic drugs into the compounding system. This can be found on page 6 of 13, paragraph 4.2, lines 189-194.

Concerning the residual levels of AD in the compartment, this is a far comment from the purpose of our study, i.e. to demonstrate the effectiveness of ozone for disinfection in automated systems used in hospital environments for compounding antineoplastic drugs.

Comment 12. Comment the limitations of the study, such as low number of replicates.

Response 12:

Thank you for pointing this out. We agree with this comment. Therefore, in the body of the text, the average values of the results obtained were given. However, the execution of the test in triplicate for the two different ozone concentrations was specified and summary tables with results and standard deviation were included. A detailed overview of the tests performed can be found in Table 1 and Table 2 (page 4 of 13, paragraph 2, lines 141-148).

Comment 13. Page 8, line 355 4-log microbial “reduction “ is missing.

Response 13: Thank you for pointing this out. We agree with this comment. Therefore, we have added the word ‘reduction’ (page 6 of 13, paragraph 3, lines 174-175).

Reviewer 2 Report

Comments and Suggestions for Authors

I read with interest the paper titled "Use of ozone for disinfection of automatic system for antineoplastic compounding". The paper is good, however, I'm sending minor comments that could enhance the manuscript. 

1. The entire paper is based on a specific equipment. This information should be present in the title, since this way, the reader is led to believe that the article is about the ozone technique, when it is mostly based on this technique in this specific equipment. I suggest something like "Use of ozone for disinfection of  PHARMODUCT® automatic dispensing system for antineoplastic compounding"

2. This is an innovative method for sterilization, and it should be the focus of the paper. I believe the paper could be enhanced if this technique could be discussed as an opportunity for different equipment other than the presented. 

3. The equipment-technique pair seems to have a good result on the microorganisms after compounding, which is quite impressive. How this enhances the sterelization with other techniques? Whats the rates of contamination with other techniques? Could you please briefly describe the advantages (and also advantages, comparing to common techniques for sterilization?)

4. From the practical point of view to view, I would like to see results of contamination with other substances rather than microorganisms with this technique of sterilization and equipment. For eg. after compounding cyclophosphamide (or other antineoplastic agents). Is gas spreading antineoplastic substances within the the equipment?

5. What the impact for the equipment that ozone desinfection have? Is it something that degrades components of the equiment?

Author Response

Comment 1: The entire paper is based on a specific equipment. This information should be present in the title, since this way, the reader is led to believe that the article is about the ozone technique, when it is mostly based on this technique in this specific equipment. I suggest something like "Use of ozone for disinfection of  PHARMODUCT® automatic dispensing system for antineoplastic compounding"

Response 1: Thank you for pointing this out. We agree with this comment. Therefore, we have corrected the title of the article, as per your instructions (page 1 of 13, Title of paper, line 2).

Comment 2. This is an innovative method for sterilization, and it should be the focus of the paper. I believe the paper could be enhanced if this technique could be discussed as an opportunity for different equipment other than the presented.

Response 2: Thank you for pointing this out. We agree with this comment. Therefore, we have taken up your suggestion with interest and have included the reference to the applicability of the bio-decontamination process (page 1 of 13, paragraph “Abstract”, lines 19-20 and page 11 of 13, paragraph 5, lines 431-433).

Comment 3. The equipment-technique pair seems to have a good result on the microorganisms after compounding, which is quite impressive. How this enhances the sterilization with other techniques? What’s the rates of contamination with other techniques? Could you please briefly describe the advantages (and also advantages, comparing to common techniques for sterilization?)

Response 3: Thank you for pointing this out. We agree with this comment. Therefore, we have better developed the section on the advantages and disadvantages between manual cleaning and the use of ozone (page 5-6 of 13, paragraph 3, lines 157-173).

However, the focus has been on manual disinfection techniques and the chemical disinfection agents used (e.g. chlorine derivatives, quaternary ammonium salts and alcohols) applied to automated systems for the preparation of antineoplastic drugs in hospitals, and not on all commercially available disinfection techniques.

Comment 4. From the practical point of view to view, I would like to see results of contamination with other substances rather than microorganisms with this technique of sterilization and equipment. For eg. after compounding cyclophosphamide (or other antineoplastic agents). Is gas spreading antineoplastic substances within the equipment?

Response 4: Thank you for pointing this out. Although the comment is not pertinent to the aim of the article, there is no risk that ozone may contribute to the spread of antineoplastic agent residues because the biodecontamination cycle involves:

  • Initial ozonisation phase at 40 ppm for 40 minutes
  • Ozone removal in the chamber down to 5 ppm, with opening of the chemical kerulite filter
  • Ventilation phase for 5/10 minutes
  • Start-up of activities

However, during the compounding activities, there is a unidirectional laminar flow (LAF) (air velocity = 0.45 m/sec ± 20%), which ensures a continuous grade A air wash and thus the previously achieved aseptic conditions, guaranteeing the safe handling of the drugs.

Comment 5. What the impact for the equipment that ozone disinfection have? Is it something that degrades components of the equipment?

Response 5: Thank you for pointing this out. We agree with this comment. Therefore, We have specified this point in paragraph 5 ‘Conclusions’ (page 11 of 13, paragraph 5, lines 423-425). As widely demonstrated in the literature, ozone has a high compatibility with various materials and polymers commonly used in the pharmaceutical industry, so there is no risk of degradation of equipment components.

Reviewer 3 Report

Comments and Suggestions for Authors

The authors states that the scope of this study was to demonstrate the ozone efficacy for disinfection of the PHARMODUCT® automatic dispensing system, before starting the antineoplastic preparation, ensuring a higher grade of cleanliness. The manuscript is more as an advertise for the product presented, anf less on the scientific and research part. I recomand resubmission after some research studies are conducted with this promising device.

Reject with resubmission.

Author Response

Comment 1: The authors states that the scope of this study was to demonstrate the ozone efficacy for disinfection of the PHARMODUCT® automatic dispensing system, before starting the antineoplastic preparation, ensuring a higher grade of cleanliness. The manuscript is more as an advertise for the product presented, anf less on the scientific and research part. I recommend resubmission after some research studies are conducted with this promising device.

Thanking you for your interest in reading this scientific article, however, we would like to emphasise that, although some experimental aspects can be further investigated and deepened, the results obtained may constitute a good starting point for the development of a biodecontamination system with ozone, tested on the PHARMODUCT system but actually applicable to all automated compounding systems on the market.

Round 2

Reviewer 1 Report

Comments and Suggestions for Authors

In my opinion the authors addressed all of my general and specific questions and revised the manuscript accordingly so it is now ready for publication.

Reviewer 3 Report

Comments and Suggestions for Authors

The revised manuscript is an improvement over the original submitted manuscript. In its current form, it is being considered for publication in the journal Pharmaceuticals.